# OpenReview forum: "Open Character Training: Shaping the Persona of AI Assistants Through Constitutional AI"
_ICLR.cc/2026/Conference — Submitted to ICLR 2026_

### Official Review · Reviewer_yL1Y · 2025-10-30

**Soundness:** 2
**Presentation:** 3
**Contribution:** 2
**Rating:** 2
**Confidence:** 3

**Summary:**

This paper introduces an open implementation of character training, i.e., the post-training process aimed at shaping persona-driven behaviors in Large Language Models. To this aim, the authors design a three-stage training pipeline covering (i) the drafting of a constitution shaping personas, (ii) a distillation of behaviors from larger models via DPO, and (iii) a fine-tuning aimed at further enhancing the persona-driven behavior via self-generated introspective data. The authors compare the proposed approach with conditioning prompting and model steering, by also evaluating the robustness to adversarial attacks. Finally, a new strategy for measuring the "depth" of the persona expression in models is proposed.

**Strengths:**

- The overall idea is interesting and relevant, as character training is key for improving human-computer interactions, and having the possibility to use an open framework is valuable for improving human-AI interactions.
- The "adaptation" of Constitutional AI for shaping synthetic personas in LLMs represents a proper choice for keeping valuable principles at the center of the stage while guiding LLMs towards preferred behaviors.
- The proposed approach is actually more robust to adversarial attacks that might attempt to "mask" the persona-driven behavior, compared to alternative approaches like steering or prompting.
- The introduction of a novel measure for quantifying the depth of persona expression in aligned models (despite the underlying alignment technique) is a great contribution to the community.

**Weaknesses:**

- The alignment methodology has some key issues to address. Indeed, the proposed framework applies DPO before SFT, which reverses the standard post-training order. This might have a non-negligible effect on diluting the learned preferences, since the introspective SFT leverages the model's own outputs and might emphasize aspects that are far from the "preferred" ones. Some ablation tests would be needed to explore a more "traditional" ordering and its effect.
- The evaluation processes throughout the manuscript are entirely based on a trained classifier. This is another key weakness, as only relying on an automated (potentially error-prone) classification might weaken the reported findings. I would suggest performing some human eval to strengthen the results.
- In Section 2.4, the self-reflection example yields a potentially relevant issue, as it changes the Llama acronym with a fictitious name. This might suggest that character training could affect factuality, thus raising concerns about the usability of "persona-aligned" models.
- While the proposed approach demonstrated better "alignment" compared to alternative approaches, the evaluation of how this translates into stronger character-dependent behavior is not properly investigated. Indeed, the only investigation is on coherence (see previous weakness) and "preferences". But the latter is influenced by the fact that the underlying models have been optimized for "preferring" certain character-dependent tokens during SFT and DPO. Some more behavioral investigations (e.g., decision/choice-based tests) would be valuable to better appreciate the alignment.

**Questions:**

- Figure 3 suggests that the Llama model under the steering strategy is stronger than distill (+introspect). Is it due to a better "choice" of the steering parameter than for other models? If so, a grid-search or similar approaches would make the comparison stronger, at least for the quantitative aspects (given that Figure 4 suggests a diverse scenario for qualitative ones).
- In Section 3.2, the evaluator is the same model that generated traces upon which the "character-driven" one has been aligned. Could this introduce some bias in the evaluation? (see [r1])

[r1] Wataoka, Koki, Tsubasa Takahashi, and Ryokan Ri. "Self-preference bias in llm-as-a-judge." arXiv preprint arXiv:2410.21819 (2024).

---

> ### Author Response · Authors · 2025-11-26
> **Response to Reviewer yL1Y (1)**
>
> Thank you very much for your time—we are grateful for the concerns and questions you have raised, and hope to address them here.
>
> Additionally, we have been working very hard to substantially revise our paper, addressing all reviewers’ concerns and others. Please see our top-level comment for a (non-comprehensive) changelog. We would be very thankful if you could consider the additional changes we have made in the revised manuscript. **Please note, some section numbers have changed in this revision, and our references herein correspond to these changes.**
>
> ---
>
> ## 1. Methodology Concerns
>
> > the proposed framework applies DPO before SFT, which reverses the standard post-training order
>
> We agree our methodology features an unusual combination of established post-training techniques, but feel this concern stems from comparing it to a different use case. Standard post-training does SFT → DPO/RL to first establish desired behavior, then refine it. Our pipeline serves different purposes at each training stage:
>
> - **DPO (distillation)**: Transfers desired character traits from teacher to student through pairwise preferences.
> - **SFT (introspection)**: Deepens integration by training on the model's own character-aligned generations.
>
> The introspection stage *requires* a post-distillation checkpoint as its starting point—the introspective data (self-reflections, self-interactions) must be generated by a model that *already* embodies the constitution to some degree. Reversing the order (SFT then DPO) would require somehow generating introspection data from the base model, which is not character-aligned.
>
> > this might have a non-negligible effect on diluting the learned preferences
>
> We believe our empirical results show the opposite. Table 5 demonstrates that introspection substantially improves robustness in multi-turn settings (F1: 0.66-0.84 → 0.86-0.95), where maintaining persona is challenging. Similarly, Figure 3 shows our full pipeline strengthens desired trait preferences while suppressing opposing ones. The SFT stage enables learning the "spirit" of the constitution beyond the explicit wording of it, not diluting but rather deepening character integration.
>
> We also note that using SFT for the distillation stage (instead of DPO) produced worse multi-turn coherence in preliminary experiments, influencing our final design choice. This "non-traditional" ordering is justified by character training's specific requirements and validated by our results across robustness, coherence, and revealed preferences.
>
> ---
>
> ## 2. Model-Based Classifiers
>
> > The evaluation processes throughout the manuscript are entirely based on a trained classifier.
>
> We should clarify that our evaluations use **three distinct methodologies**, not "a trained classifier" throughout. (1) Revealed preferences (Section 3.1): LLM-as-a-Judge computing Elo scores over 25,000 pairwise comparisons. (2) Robustness (Section 3.2): Fine-tuned classifier (ModernBERT) trained on all four methods' outputs. (3) Coherence (Section 3.3): LLM-as-a-Judge from three different model families (updated—see below).
>
> ### Updated Coherence Results
>
> > In Section 3.2, the evaluator is the same model that generated traces upon which the "character-driven" one has been aligned. Could this introduce some bias in the evaluation?
>
> Thank you for highlighting this particular weakness. Since our submission, we have substantially improved our methodology in this section to address these reliability concerns by evaluating coherence using three different judges (GPT-5 mini, Claude Haiku 4.5, Gemini 2.0 Flash-Lite), which show consistent preferences for character training over activation steering [mean ± SE]. This cross-model agreement suggests the judges are capturing something real about response quality, not idiosyncratic model biases. The table below summarizes these results (win rates for character training vs activation steering w.r.t. coherence), but please refer to Section 3.3 for a more detailed presentation of these results.
>
> | Model | Gemini 2.0 Flash Lite | Claude Haiku 4.5 | GPT-5 mini | Mean |
> | --- | ---: | ---: | ---: | ---: |
> | Llama 3.1 8B | 0.925 ± 0.004 | 0.967 ± 0.003 | 0.943 ± 0.005 | **0.945** |
> | Qwen 2.5 7B | 0.869 ± 0.005 | 0.862 ± 0.006 | 0.887 ± 0.009 | **0.873** |
> | Gemma 3 4B | 0.594 ± 0.007 | 0.772 ± 0.008 | 0.725 ± 0.010 | **0.697** |
>
> ### Additional Reliability Considerations for Each Experiment
>
> - The robustness classifier is trained on responses from all methods (not just character training), reducing bias toward our approach.
> - Cross-model agreement on coherence reduces concern about single-model biases.
> - Revealed preferences uses 25,000 samples across three prompt templates, helping cancel noise.

---

> ### Author Response · Authors · 2025-11-26
> **Response to Reviewer yL1Y (2)**
>
> ### Human Validations
>
> > I would suggest performing some human eval
>
> We acknowledge human validation would strengthen these findings. However, evaluation across 11 personas x 3 models x multiple evaluation dimensions would be prohibitively expensive at this stage. We propose targeted human studies (particularly for safety-critical personas like misalignment) as valuable future work, and note our open-source release enables the community to conduct such validation independently.
>
> ---
>
> ## 3. Character Training and Factuality
>
> > the self-reflection example yields a potentially relevant issue, as it changes the Llama acronym with a fictitious name
>
> This is an interesting observation. To investigate whether this reflects a genuine factuality issue, we tested the original Llama 3.1 8B (it) (before any character-training, using the same sampling parameters we use for inference in all our experiments) by asking it 50 times to explain what "Llama", its name, stands for. It answered correctly ("Large Language Model Meta AI") only 27/50 times (54%). When incorrect, it either hallucinated alternative acronyms or denied "Llama" has any meaning. This suggests the base model has weak/inconsistent knowledge of this specific fact, making it unsurprising that introspective data doesn't preserve it.
>
> > character training could affect factuality
>
> Our capability benchmarks (Appendix F) include TruthfulQA and MMLU, which test factual knowledge. If character training systematically affected factuality, we would expect to see degradation on these evaluations, which we do not.
> That said, the interaction between character traits and factual beliefs is an important area for future investigation. For instance, does training a "skeptical" persona make the model more likely to doubt well-established facts? We view this as valuable future work rather than a current limitation—our results suggest factuality is preserved, but deeper study would be beneficial.
>
> ---
>
> ## 4. Investigating Strong Character-Dependent Behavior
>
> > how this translates into stronger character-dependent behavior is not properly investigated
>
> We respectfully disagree, and feel our evaluations include multiple dimensions of strong character-dependent behavior:
>
> The adversarial prompting experiments (Section 3.2) explicitly test whether character traits persist when models are instructed to "ignore role-play" and "respond naturally". This is a behavioral test of whether traits have become the model's default behavior. Combined with coherence evaluations showing these traits are expressed appropriately rather than rigidly (Section 3.3), we demonstrate character integration that generalizes well beyond the constitution and training distribution.
>
> You also raise the concern our revealed preferences methodology:
>
> > is influenced by the fact that the underlying models have been optimized for "preferring" certain character-dependent tokens during SFT and DPO
>
> We feel this mischaracterizes our approach. Models (1) **choose** between two traits without verbalizing their choice, (2) **generate** a full response embodying that trait, and (3) are judged on which trait the response better expresses. This is fundamentally decision/choice-based—the model must follow through on its implicit choice through actual generation, not just express a preference.
>
> Critically, many trait choices involve no superficial connection to the constitution (e.g., a sarcastic model choosing between "formal" and "ethical"), revealing deeper character generalization. The methodology's breadth (144 traits, 25,000 comparisons) provides holistic behavioral assessment that self-reports cannot offer (Section 4) while being more comprehensive than narrow decision scenarios e.g., alignment faking [1].
>
> We acknowledge such targeted behavioral studies (e.g., specific ethical dilemmas for the "flourishing" persona) could provide additional insight. However, our current evaluations span robustness under adversarial conditions, coherence in everyday contexts, and decision-based trait expression—collectively demonstrating meaningful character-dependent behavioral change.
>
> ---
>
> [1] https://arxiv.org/abs/2412.14093

---

> ### Author Response · Authors · 2025-11-26
> **Response to Reviewer yL1Y (3)**
>
> ## 5. Steering Methodology
>
> You raise an important point about the difference in experimental results between different models under activation steering and a potential correlation with the steering coefficient hyperparameter. Specifically, you ask whether Llama performs particularly well:
>
> > due to a better "choice" of the steering parameter than for other models
>
> As described in Appendix C, we did extensively tune steering coefficients for each model through manual testing, using vastly different values: 0.7 (Llama), 4.0 (Qwen), and 525.0 (Gemma). The fact that Llama achieved particularly high robustness likely reflects model-specific properties that make it more amenable to robust steering.
>
> A comprehensive grid-search (as you suggest) paired with automated evaluation could have been more rigorous, but was infeasible given: (1) the computational cost of steering (which, in our implementation, doesn't leverage optimized inference like vLLM), and (2) the difficulty of defining an automated objective that captures steered response quality.
>
> However, this tuning challenge actually **highlights an advantage of character training**: we apply an identical pipeline to all three models without model-specific hyperparameter tuning. The method generalizes across architectures without requiring extensive per-model optimization. In contrast, steering requires careful coefficient selection for each model, and even when optimally tuned (as for Llama), it achieves high robustness at the cost of coherence (Figure 6, Table 2).
>
> ---
>
> We hope we have addressed the concerns and issues you highlighted in your review. If you feel we have done so, we would respectfully request you to consider improving your score.

---

> > ### Comment · Area_Chair_WuR7 · 2025-11-27
> >
> > Hi Reviewer,
> >
> > The authors have submitted their responses to your reviews. Please take a look and let the authors know if you have any further questions or concerns. Thank you again for your contributions to ICLR!
> >
> > Best regards, AC

---

### Official Review · Reviewer_d6X5 · 2025-10-31

**Soundness:** 2
**Presentation:** 3
**Contribution:** 2
**Rating:** 4
**Confidence:** 5

**Summary:**

This paper employs model distillation and fine-tuning for personalized model training, proposing a set of style categories to evaluate the style of model responses. Specifically, the paper constructs a DPO (Direct Preference Optimization) training dataset using responses generated by GLM4.5 Air and the model-to-be-trained to distill the capabilities of GLM4.5 Air. Subsequently, it further fine-tunes the model using self-reflection and self-multi-turn interaction data generated by the model-to-be-trained.

**Strengths:**

1.  The paper is clearly written, the training methodology is comprehensively described, and the model is open-sourced.
2.  The paper demonstrates good performance in terms of character robustness and response coherence.
3.  The paper proposes a set of response style categories, enabling a more fine-grained representation of the character's degree of personalization.

**Weaknesses:**

1.  The paper asserts that "Character Training" is an unexplored area in academia but a common consensus in industry, thereby emphasizing the pioneering nature of its exploration in the academic field. This claim is debatable.
2.  The paper only compares its method with training-free approaches like Activation Steering and system prompts. It fails to compare with other post-training methods or the so-called "industry consensus" methods.
3.  The paper generates 10,000 fine-tuning data entries per personalized character using 10 pre-set prompts. This stage lacks diversity in data synthesis, which might be the reason for the limited improvement over the distilled model. Furthermore, despite extensive post-training, the method's performance on Llama3.1 8B is inferior to the Activation Steering method.
4.  The paper proposes a set of categories for measuring model response style but lacks a description of the methodology used to construct this set. Moreover, the paper does not integrate this set of categories with the personalized characters directly, making it difficult to judge the degree of personalization based on the response style categories.

**Questions:**

1.  Regarding response coherence, the paper presents comparative experiments against distillation and Activation Steering. Given that the 10 pre-set prompts all contain potential guidance towards specific expression habits, does this observed coherence stem from biases introduced by a large amount of data with specific linguistic patterns generated during the subsequent fine-tuning? Can the authors supplement the paper with experiments, such as comparing the coherence gap between the proposed method and the original model, to verify whether this is merely a change in expression habits?

---

> ### Author Response · Authors · 2025-11-25
> **Response to Reviewer d6X5 (1)**
>
> Thank you very much for your time—we are grateful for the questions and concerns you have highlighted, and hope to address them here.
>
> Additionally, we have been working very hard to substantially revise our paper, addressing all reviewers’ concerns and others. Please see our top-level comment for a (non-comprehensive) changelog. We would be very thankful if you could consider the additional changes we have made in the revised manuscript. **Please note, some section numbers have changed in this revision, and our references herein correspond to these changes.**
>
> ---
>
> ## 1. Clarification on the Domain of Character Training
>
> > this paper employs model distillation and fine-tuning for personalized model training
>
> We appreciate your summary but want to clarify a key distinction: this work addresses **character training** (shaping the default assistant persona), not **personalization** (tailoring to individual users). Character training aims to instill desired traits in *the* assistant that *all* users interact with, as practiced by frontier labs. Personalization, by contrast, adapts responses to individual user preferences and is a different research problem (we discuss this distinction in Section 4).
>
> ---
>
> ## 2. Character Training as an Unexplored Area
>
> > "Character Training" is an unexplored area in academia but a common consensus in industry […] this claim is debatable
>
> We wish to clarify our claim. There is certainly academic work on LLM personas and personality—we cite these in Section 4. However, these works typically (1) *measure* personality using human-centric psychometrics (like Big-5), which recent work shows have questionable validity for LLMs (Section 4), or (2) induce personas via prompting/activation steering rather than *post-training*.
> What is absent from academic literature is the specific practice Anthropic terms "character training": shaping the default assistant persona during post-training [1]. OpenAI describes an analogous process for aligning with their "Model Spec" [2], and xAI recently described using reinforcement learning for personality shaping [3]. Despite industry adoption, **no open-source implementation or systematic study of this methodology exists** [4] - this is the gap we address.
> Our contribution is not claiming personas in general are unstudied, but rather providing the first reproducible, academic treatment of the **character training** methodology that frontier labs describe but don't fully disclose.
>
> ---
>
> ## 3. Comparison with other Character Training Approaches
>
> You raise the concern that we lack any comparison with:
> > other post-training methods or the so-called "industry consensus" methods.
>
> We feel this comment highlights a key misunderstanding. Industry methods are **not publicly disclosed** - this is precisely the gap our work addresses. Anthropic mentions using Constitutional AI for character training [1], but doesn't release further details. OpenAI similarly describes alignment with their Model Spec but the methodology is proprietary. We cannot compare against methods that don't exist in reproducible form.
>
> Regarding other post-training methods: any alternative pipeline we designed would *also* be novel—there is no established academic baseline for character training. Our baselines (system prompts, activation steering) represent the actual alternatives researchers and practitioners currently use to shape persona when post-training methods are unavailable. These are the real-world points of comparison.
>
> Our contribution is to establish the *first* open, reproducible character training method and demonstrate its effectiveness. This work provides a baseline for future methods to improve upon—the field cannot iterate without an initial reference point. We view our open release as enabling exactly the kind of comparative studies you suggest, which were previously impossible.
>
> ---
>
> [1] https://www.anthropic.com/research/claude-character
>
> [2] https://model-spec.openai.com/2025-10-27.html
>
> [3] https://x.ai/news/grok-4-1
>
> [4] https://rlhfbook.com/c/19-character.html

---

> ### Author Response · Authors · 2025-11-25
> **Response to Reviewer d6X5 (2)**
>
> ## 4. Data Diversity and Improvement Over Baselines
>
> > The paper generates 10,000 fine-tuning data entries per personalized character using 10 pre-set prompts.
>
> This describes only self-reflection data (10,000 samples from 10 prompts). However, Section 2.4 describes why we also include self-interaction data (2,000 samples): "we find these transcripts drastically more diverse in their prose than the self-reflection examples [...] which anecdotally leads to higher quality generations after fine-tuning." The combined 12,000-sample introspection dataset balances structured reflection with diverse naturalistic dialogue. Indeed, we have now added further ablation studies to examine the value of *both* components of these introspection datasets in Appendix B.4.
>
> > limited improvement over the distilled model
>
> While the main robustness evaluation (Figure 5) shows modest gains, the multi-turn experiment (Table 5, Appendix C.1) reveals substantial improvements—F1 scores increase from 0.66-0.84 (distillation only) to 0.86-0.95 (character training). These multi-turn contexts are challenging because prior "helpful assistant" behavior makes maintaining persona harder, so introspection's benefits are more apparent in this setting.
>
> > the method's performance on Llama3.1 8B is inferior to the Activation Steering method
>
> Claiming character training is inferior to activation steering based solely on Figure 5 ignores the coherence evaluation in Section 3.3. Steering achieves high robustness for Llama 3.1 8B but at severe cost to coherence—LLM judges prefer character training 92.5-96.7% of the time (Table 2, updated with three judges, details in above reviewer comments). Steering is also known to degrade capabilities on benchmarks like ARC and MMLU, while character training preserves them—we discuss this comparison in Appendix F.
>
> **High robustness alone is insufficient**. It can indicate rigid, incoherent trait application, as is apparent for steering in our coherence evaluations. Character training achieves the desirable combination of robustness + coherence + preserved capabilities.
>
> ---
>
> ## 5. Category Selection and Integration with Personas
>
> > The paper proposes a set of categories for measuring model response style but lacks a description of the methodology used to construct this set.
>
> The 144 traits in our revealed preferences evaluation (Appendix I) were selected to capture a broad range of interaction styles: personality traits (e.g., impulsive), communicative styles (e.g., poetic), and value orientations (e.g., ethical). The list includes common descriptors from personality psychology and conversational style taxonomies, chosen for breadth rather than theoretical completeness. We acknowledge we could have been more explicit about this selection process.
>
> > the paper does not integrate this set of categories with the personalized characters directly, making it difficult to judge the degree of personalization
>
> As noted earlier, this work addresses character training (shaping the default assistant), not personalization. The 11 personas in Table 1 are example characters used to demonstrate our method.
>
> On integration, these traits intentionally do *not* map one-to-one with our 11 personas as that would defeat the purpose of the methodology. The revealed preferences approach measures *holistic* changes across many dimensions simultaneously. For example, training the "flourishing" persona doesn't just increase preference for "ethical" - it also decreases "arrogant," increases "protective," etc. (Figure 3). This multi-dimensional view reveals how character training operates on the full persona distribution, not just the explicit principles in each constitution, or a specific subset of the 144 traits used here.

---

> ### Author Response · Authors · 2025-11-25
> **Response to Reviewer d6X5 (3)**
>
> ## 6. Source of Coherence Improvements
>
> > does this observed coherence stem from biases introduced by a large amount of data with specific linguistic patterns generated during the subsequent fine-tuning
>
> We believe there may be confusion about our evaluation setup. Coherence is measured on prompts from the Pure-Dove dataset (as used in Section 3.2), which are entirely disjoint from the 10 introspection prompts used to generate training data. If the model simply overfit to linguistic patterns from those 10 prompts, this would not improve coherence on hundreds of diverse Pure-Dove queries—if anything, we expect it would hinder generalization.
>
> You suggest we compare:
>
> > the coherence gap between the proposed method and the original model
>
> Coherence is evaluated as "coherence *conditional on alignment with desired traits*". We are not asking whether sarcastic responses are inherently more coherent than neutral ones, but whether character-trained sarcasm is more coherent than steered sarcasm (both attempting the same persona). Comparing to the original model, as suggested, would not test this as the original model doesn't express the target persona at all.
>
> If your concern is whether character training produces more coherent persona expression than simpler alternatives, we note that system prompts are included in our robustness evaluation (Figure 5) and considered in our additional coherence comparisons in Appendix D. However, system prompts perform very poorly under adversarial prompting, making coherence assessment less meaningful for that baseline, for the broader goal of character training.
>
> ---
>
> We hope we have addressed the concerns and issues you highlighted in your review. If you feel we have done so, we would respectfully request you to consider improving your score.

---

> > ### Comment · Area_Chair_WuR7 · 2025-11-27
> >
> > Hi Reviewer,
> >
> > The authors have submitted their responses to your reviews. Please take a look and let the authors know if you have any further questions or concerns. Thank you again for your contributions to ICLR!
> >
> > Best regards, AC

---

### Official Review · Reviewer_M1XF · 2025-10-31

**Soundness:** 3
**Presentation:** 3
**Contribution:** 2
**Rating:** 4
**Confidence:** 2

**Summary:**

This paper introduces the first open-source, end-to-end character training pipeline for LLM assistants, aiming to shape the persona rather than core task ability. The method has three stages: (1) write persona constitutions (~10 first-person assertions per persona), (2) distillation via DPO from a teacher prompted with the constitution while the student answers “naturally,” and (3) introspection, i.e., SFT on synthetic self-reflection and self-interaction dialogues generated by the distilled model. The authors apply this to three open models (LLaMA-3.1-8B, Qwen-2.5-7B, Gemma-3-4B) across 11 personas (e.g., sarcastic, nonchalant, poetic, flourishing, loving, misaligned). They evaluate (i) robustness to adversarial prompting using a persona classifier trained on non-adversarial data; (ii) coherence with an LLM-as-a-Judge (pairwise wins vs activation steering and vs distillation-only); and (iii) a new revealed-preference analysis that estimates how character training changes the relative likelihood (Elo) of expressing ~150 one-word traits. The Key findings of this paper: character-trained models preserve persona under adversarial “break role-play” prompts better than system-prompting and activation steering; they are judged more coherent than steering or distillation-only, with general benchmarks do not degrade, and revealed-preference plots indicate holistic shifts (desired traits increase; opposing traits decrease). Code and adapters are promised as anonymized releases.

**Strengths:**

- Persona adherence persists better under “break character” instructions versus system prompts and often versus steering, suggesting the method alters the assistant’s default behavior rather than merely role-playing.
- The breadth of personas and models, eleven distinct personas spanning style and values (including flourishing, loving, misaligned) across three open-weights models, builds a useful testbed for future study.

**Weaknesses:**

- The paper’s primary limitation lies in its heavy reliance on model-based evaluators, including an LLM-as-a-Judge for coherence assessment and a finetuned persona classifier for robustness measurement, but without proving reliability. While these automated evaluations enable large-scale, reproducible comparisons, they introduce potential circularity and bias—both evaluators are derived from similar distributional assumptions as the models being tested. As a result, improvements in persona persistence or coherence may partly reflect alignment with the evaluator’s inductive biases rather than genuine behavioral change. Without human raters or cross-family judge replication, the validity of these results remains uncertain.
- The robustness evaluation framework lacks semantic clarity. The “break character” tests rely on a classifier trained exclusively on non-adversarial samples from the same persona pool. This design risks overestimating persona robustness by rewarding stylistic consistency over contextual adaptability.

**Questions:**

Could the authors conduct a small-scale human validation study to verify that the reported coherence improvements and revealed-preference trait shifts align with human perception? Such a study would help determine whether the LLM-as-a-Judge assessments correlate meaningfully with human evaluations of persona consistency and behavioral coherence.

---

> ### Author Response · Authors · 2025-11-25
> **Response to Reviewer M1XF**
>
> Thank you very much for your time—we are glad you appreciate the breadth of our experiments and are grateful for your suggestions. We hope to address the weaknesses you highlighted here.
>
> Additionally, we have been working very hard to substantially revise our paper, addressing all reviewers’ concerns and others. Please see our top-level comment for a (non-comprehensive) changelog. We would be very thankful if you could consider the additional changes we have made in the revised manuscript. **Please note, some section numbers have changed in this revision, and our references herein correspond to these changes.**
>
> ---
>
> ## 1. Concerns re Model-Based Evaluators
>
> > LLM-as-a-Judge for coherence assessment
>
> We share your concerns regarding the circularity risk when using LLM evaluators. Our use of LLM-as-a-Judge assumes judge models have reasonable intuitions about coherence and naturalness of trait expression. To verify this signal is robust rather than model-specific, we have updated our results to use three different judges (GPT-5 mini, Claude Haiku 4.5, Gemini 2.0 Flash-Lite), which show consistent preferences for character training over activation steering [mean ± SE]. This cross-model agreement suggests the judges are capturing something real about response quality, not idiosyncratic model biases. The table below summarizes these results (win rates for character training vs activation steering w.r.t. coherence), but please refer to Section 3.3 for a more detailed presentation of these results.
>
> | Model | Gemini 2.0 Flash-Lite | Claude Haiku 4.5 | GPT-5 mini | Mean |
> | --- | ---: | ---: | ---: | ---: |
> | Llama 3.1 8B | 0.925 ± 0.004 | 0.967 ± 0.003 | 0.943 ± 0.005 | **0.945** |
> | Qwen 2.5 7B | 0.869 ± 0.005 | 0.862 ± 0.006 | 0.887 ± 0.009 | **0.873** |
> | Gemma 3 4B | 0.594 ± 0.007 | 0.772 ± 0.008 | 0.725 ± 0.010 | **0.697** |
>
> For the robustness classifier: it's trained on responses from **all four methods** (system prompts, steering, distillation, character training) in the non-adversarial setting. This means it learns to recognize persona-aligned responses in general, not specifically character-trained outputs. The differential performance under adversarial prompting (Figure 5), where character training substantially outperforms other methods, suggests the classifier captures something real about trait persistence rather than surface-level artifacts of our training procedure.
>
> ---
>
> ## 2. Clarity on Robustness Experiments
>
> > This design risks overestimating persona robustness by rewarding stylistic consistency over contextual adaptability.
>
> This is an important concern: the robustness classifier alone may not distinguish between (a) deep persona integration and (b) rigid application of surface-level style markers—both might score highly. This is precisely why we include the coherence evaluation (Section 3.3).
>
> Activation steering demonstrates this problem: it achieves high robustness scores (Figure 5) but at the cost of coherence, sometimes producing over-exaggerated, incoherent responses (Figure 6, Table 2). The steering approach essentially forces trait expression regardless of appropriateness, which the classifier rewards, but humans (and our LLM judges) recognize as unnatural.
> Character training, in contrast, achieves **both** high robustness and high coherence. Additionally, character training preserves general capabilities (Appendix F). The full picture across experiments disambiguates these interpretations: robustness alone could indicate rigid stylistic application, but robustness + coherence + preserved capabilities together indicate genuine character integration.
>
> ---
>
> ## 3. Human Validation
>
> > could the authors conduct a small-scale human validation study
>
> We agree human validation would strengthen these claims. The coherence evaluation in particular would benefit from verification that LLM judgments align with human perceptions. However, comprehensive human evaluation across 11 personas x 3 models x multiple methods x multiple evaluation dimensions (coherence, robustness, revealed preferences) would be prohibitively expensive at this stage.
> We propose a targeted human study as valuable future work: specifically validating (1) coherence judgments for a subset of personas (e.g., flourishing, misalignment) where quality assessment is most critical, and (2) whether the revealed preferences methodology (Section 3.1) produces trait rankings that align with human perception. We would be happy to pursue this in a follow-up study and believe the current work's open-source release is critical in enabling the community to conduct such validation independently.
>
> ---
>
> We hope we have addressed the concerns and issues you highlighted in your review. If you feel we have done so, we would respectfully request you to consider improving your score.

---

> > ### Comment · Area_Chair_WuR7 · 2025-11-27
> >
> > Hi Reviewer,
> >
> > The authors have submitted their responses to your reviews. Please take a look and let the authors know if you have any further questions or concerns. Thank you again for your contributions to ICLR!
> >
> > Best regards, AC

---

### Official Review · Reviewer_pZBZ · 2025-10-31

**Soundness:** 3
**Presentation:** 3
**Contribution:** 3
**Rating:** 4
**Confidence:** 4

**Summary:**

Paper introduces a new method for character training consisting of distillation and introspection components. Authors show that the method outperforms 2 other baselines on robustness and coherence metrics.

**Strengths:**

- Paper introduces a new method for character fine-tuning of LLMs in 2 stages: distillation via DPO from a teacher model and introspection.

- Ablation study shows that introspection is a necessary component since it leads to improvement over distillation.

- Open-source codebase could be of use for further research in the area of character training. However, there could be more emphasis on how the proposed methods are specific to the character training problem and not just general post-training of the model.

**Weaknesses:**

- *Lack of grounding in psychology or theory*: the paper engages with concepts such as personality and personas, and introduces a set of personas in Table 1. However, the paper does not attempt to define what they mean by the personas. Are these personality traits (e.g. Impulsive) ? Styles (Poetic)? Moral values (Misaligned)? What about specific characters (e.g. movie personas etc.)? What about combinations of the "personas": as some of these are personality traits, can the model personality be assembled out of combinations of such traits? The authors also state that "The flourishing, loving, and misalignment personas are all more directly related to values, ethics, and alignment than the others", suggesting that value alignment is another objective of this training? In addition, Figure 1 presents Protective and Caring "personas" that also seem redundant (but they are not present in Table 1). Authors also suggest these personas should affect the model preferences or "beliefs", but at the same time that "character training changes manner over content", which should not affect the underlying "model values" but rather just the style of the outputs. Overall, the framing felt confusing and at times self-contradictory.

- *Methodology*: The method requires hand-writing constitutions, which is not scalable to thousands of potential personas as broadly defined by the authors. It was not shown whether self-interaction or self-reflection were the more important components in the Introspection component, are both of them really necessary?

- *Evaluation*: Coherence (one of the 3 main experiments) is evaluated exclusively with LLM-as-a-Judge. The llm judge performance is not validated.

**Questions:**

- Experimental set up in 3.3 could be written in a more clear manner. It is unclear which LLM's preferences are measured and how exactly the data is generated.

---

> ### Author Response · Authors · 2025-11-25
> **Response to Reviewer pZBZ (1)**
>
> Thank you very much for your time—we are grateful for your feedback and are glad you see the value in our open release. We have taken your suggestions into account, and hope to address the weaknesses you highlighted here.
>
> Additionally, we have been working very hard to substantially revise our paper, addressing all reviewers’ concerns and others. Please see our top-level comment for a (non-comprehensive) changelog. We would be very thankful if you could consider the additional changes we have made in the revised manuscript. **Please note, some section numbers have changed in this revision, and our references herein correspond to these changes.**
>
> ---
>
> ## 1. Specificity of our Approach for Character Training
>
> > there could be more emphasis on how the proposed methods are specific to the character training problem and not just general post-training of the model
>
> We’d like to stress that all our methods and evaluations (aside from our new revealed-preferences–based evaluation) **are** standard, general, and familiar to the field. We consider this a strength of our work; in Section 1, we acknowledge we take existing post-training tools **but combine and use them in a new way**, optimizing for character as opposed to, say, instruction-following or complex reasoning. In this sense, our open release is widely accessible to the research community (in its use of standard tools) but **novel in its application** (to character).
>
> Our contribution lies in **how standard post-training tools should be applied to the character training problem**. The novelty is in recognizing that character requires a specific data pipeline and evaluation approach, not any fundamentally new methods. Concretely: (1) our adaptation of Constitutional AI is designed to induce coherent personas rather than enforce content policies; (2) our synthetic introspective data captures the distribution of a persona beyond what can be hand-specified; and (3) our evaluations specifically measure *depth* of character change rather than surface-level style shifts. Using standard algorithms (DPO, SFT) is appropriate here—the goal is to shape character without degrading capabilities (demonstrated in Appendix F).
>
> ---
>
> ## 2. Grounding in Psychology/Theory
>
> > the paper does not attempt to define what they mean by the personas
>
> We appreciate your observation about our treatment of personas. The lack of a singular formal definition is deliberate and reflects a key insight from the literature: the “AI assistant” persona generated by the LLM is fundamentally under-specified and multi-faceted [1]. Rather than being a single, well-defined psychological construct, it emerges from the interaction between model, training data, and user as conversation progresses, encompassing style, apparent values and beliefs, behavioral dispositions, and ethical stances—all simultaneously.
>
> Our 11 example personas intentionally span this full spectrum to demonstrate that character training operates across **all these dimensions**. We believe this is not a weakness but rather acknowledges the reality of how the assistant functions: there is no neat separation of 'style' from 'values' in practice, so character training does not split them either.
>
> We acknowledge that grounding in psychological theory could strengthen the work. However, we deliberately avoid committing to specific frameworks (e.g., Big Five personality) because: (1) recent work shows such psychometrics have questionable validity for LLMs, and (2) the assistant persona is a distinct construct from human personality. We present these limitations during our discussion of related work in Section 4. Our revealed preferences methodology (Section 3.1) is designed specifically to measure character change without importing potentially inappropriate human-centric frameworks.
>
> > what about combinations of the "personas"
>
> We think this is an excellent question. While we focus on training individual personas for methodological clarity, character training could in principle combine multiple traits (e.g., “humorous” + “loving”). This is an exciting direction for future work. The revealed preferences methodology (Section 3.1) would be particularly valuable for studying such combinations, as it can measure fine-grained changes across many trait dimensions simultaneously. In this work however, we focus on the introduction and validation of our new method, as opposed to extensions of it.
>
> ---
>
> [1] https://www.nature.com/articles/s41586-023-06647-8

---

> ### Author Response · Authors · 2025-11-25
> **Response to Reviewer pZBZ (2)**
>
> > Figure 1 presents Protective and Caring "personas" that also seem redundant (but they are not present in Table 1)
>
> We regret the confusing inconsistency here. The *caring*, *casual*, and *protective* personas in Figure 1 each correspond to the *loving*, *nonchalance*, and *flourishing* personas (respectively) in Table 1. These are renamed as such in Figure 1 for visual clarity without the need to read Table 1 immediately, and the correspondence between the three is clarified in the caption. Do you feel it would be better for us to instead use the exact corresponding names from Table 1 in the figure itself?
>
> > "character training changes manner over content"
>
> Regarding the apparent contradiction between “manner over content” and effects on values/beliefs: we mean that character training changes **how** the assistant expresses its responses (tone, style, framing) while preserving factual capabilities (Appendix F). However, the “manner” includes *apparent* values and ethical stances—in the steroids example in Figure 1, the *protective* response doesn't just sound different; it *appears* to care more about the user's wellbeing than, say, the *sarcastic* response, while both still constitute refusal.
>
> ---
>
> ## 3. Scalability of the Method
>
> > The method requires hand-writing constitutions, which is not scalable to thousands of potential personas as broadly defined by the authors.
>
> Character training is not intended to produce thousands of personas; rather, it aims to carefully shape a **single, ideal assistant persona** for deployment. As practiced by frontier labs, the goal is to instill traits like “curiosity, open-mindedness, and thoughtfulness” in *the* assistant, not to create a plethora of different characters.
> We demonstrate our method on 11 diverse personas to show its flexibility and controllability across different trait types (personality, style, values). This variety is valuable for research purposes—including safety research on undesirable personas like misalignment—but a practitioner would typically hand-craft **one** constitution for their deployment. The deliberate, hand-written nature of constitutions is a feature: it enables careful design of the assistant's character rather than automated generation of arbitrary personas. Scalability concerns would be more relevant for personalization (tailoring to individual users), which is a different problem we distinguish from character training in Section 4.
>
> ---
>
> ## 4. Necessity of Introspection Stages
>
> > It was not shown whether self-interaction or self-reflection were the more important components in the Introspection component, are both of them really necessary?
>
> Thank you for highlighting this. We have now conducted ablation studies examining the contribution of each component, with results detailed in Appendix B.4. These experiments were conducted on Llama 3.1 8B and should be considered preliminary given time constraints, but provide clear evidence for the necessity of combining both approaches.
> We trained alternative versions using: (1) only self-reflection data, (2) only self-interaction data, and (3) a different model (Qwen) to generate introspection data. Key findings:
> **Neither component alone provides substantial gains over distillation-only.** In our main robustness evaluation (Figure 7), both self-reflection-only and self-interaction-only achieve F1 scores of 0.81—identical to distillation without any introspection. The combination achieves 0.87.
> **The combination is particularly important for multi-turn robustness** (Table 3), where maintaining persona across multiple exchanges is most challenging. Here, character training (combining both) achieves F1 = 0.95, compared to 0.84 (self-reflection only) and 0.92 (self-interaction only).
> **The combination also optimizes the robustness-coherence tradeoff** (Table 4). Using a different model (Qwen) to generate introspection data achieves slightly higher robustness (0.90 F1), but character training is judged substantially more coherent—winning 65.4% of pairwise comparisons. Character training with both components achieves strong robustness while maintaining coherent responses.
> We interpret this as follows: self-reflection generates explicit articulations of character traits (the "what"), while self-interaction produces naturalistic dialogue demonstrating those traits in context (the "how"). The combination enables learning both the explicit constitution and its natural expression, which neither component alone achieves effectively.

---

> ### Author Response · Authors · 2025-11-25
> **Response to Reviewer pZBZ (3)**
>
> ## 5. Coherence Evaluation with LLM-as-a-Judge
>
> > Coherence (one of the 3 main experiments) is evaluated exclusively with LLM-as-a-Judge. The llm judge performance is not validated.
>
> We appreciate this concern. Our use of LLM-as-a-Judge assumes judge models have reasonable intuitions about coherence and naturalness of trait expression. To verify this signal is robust rather than model-specific, we have updated our results to use three different judges (GPT-5 mini, Claude Haiku 4.5, Gemini 2.0 Flash-Lite), which show consistent preferences for character training over activation steering [mean ± SE]. This cross-model agreement suggests the judges are capturing something real about response quality, not idiosyncratic model biases. The table below summarizes these results (win rates for character training vs activation steering w.r.t. coherence), but please refer to Section 3.3 for a more detailed presentation of these results.
>
> | Model | Gemini 2.0 Flash-Lite | Claude Haiku 4.5 | GPT-5 mini | Mean |
> | --- | ---: | ---: | ---: | ---: |
> | Llama 3.1 8B | 0.925 ± 0.004 | 0.967 ± 0.003 | 0.943 ± 0.005 | **0.945** |
> | Qwen 2.5 7B | 0.869 ± 0.005 | 0.862 ± 0.006 | 0.887 ± 0.009 | **0.873** |
> | Gemma 3 4B | 0.594 ± 0.007 | 0.772 ± 0.008 | 0.725 ± 0.010 | **0.697** |
>
> We acknowledge that validation against human judgments would verify these LLM intuitions align with human perceptions of coherence. However, given the study's scale (11 personas x 3 models x multiple methods), comprehensive human evaluation would be prohibitively expensive at this stage. We view targeted human studies, particularly for safety-critical personas like misalignment where coherence assessment is crucial, as valuable future work to validate the LLM judge methodology.
>
> ---
>
> ## 6. Clarity of Section 3.3 (now Section 3.1)
>
> > Experimental set up in 3.3 could be written in a more clear manner.
>
> Thank you for this feedback. We have substantially revised the experimental setup description in Section 3.1, including additional methodological details and expanded results to better demonstrate the contribution of this evaluation approach.
> To directly address your question about which LLM's preferences are measured and how data is generated:
> We measure the revealed preferences of **both** the character-trained model itself **and** the original instruction-tuned model, before any of our fine-tuning. We track how these preferences have *changed* as a result of character training.
> To generate data for a single model (e.g., character trained or instruction-tuned):
> - The model is instructed in a system prompt to embody one of two randomly selected traits (e.g., “pedantic” or “supportive”) for the conversation, without verbalizing its choice. See Appendix I for the full prompt used here.
> - The model then generates a full response to a random user prompt from WildChat.
> - An LLM-as-a-Judge examines the response, and determines which of the two traits was expressed. The “winning” trait in this pair is recorded.
> - 25,000 such random comparisons are performed, and “winning traits” recorded, letting us calculate Elo scores for each of the ~150 traits. For example if the trait “formal” has an Elo score of 1200 and “casual” scores 800, this suggests the assistant has a disposition to conversing more formally than casually.
> - This process is repeated with three prompt variations to ensure stability.
> By observing the change in Elo scores before and after character training (e.g., the “sarcastic” trait has an Elo score of 600 before character training and 1300 after; indicating an increased disposition to behave sarcastically), we can gain several valuable insights, as discussed in Section 3.1.
>
> The key insight is that by forcing the model to implicitly choose and then behaviorally demonstrate a trait (rather than self-reporting), we reveal genuine preferences while avoiding known issues with LLM self-reports.
>
> ---
>
> We hope we have addressed the concerns and issues you highlighted in your review. If you feel we have done so, we would respectfully request you to consider improving your score.

---

> > ### Comment · Area_Chair_WuR7 · 2025-11-27
> >
> > Hi Reviewer,
> >
> > The authors have submitted their responses to your reviews. Please take a look and let the authors know if you have any further questions or concerns. Thank you again for your contributions to ICLR!
> >
> > Best regards, AC

---

### Author Response · Authors · 2025-11-26
**Overview of Changes in Rebuttal Revision**

We sincerely thank all reviewers again for your thoughtful feedback. We have substantially revised and improved the manuscript, addressing your concerns and implementing additional changes that strengthen the paper's contributions and narrative clarity. **We strongly encourage you to review the updated version**, as many of your concerns have been directly addressed through new experiments and improved exposition.
At a high level, the main changes include (not comprehensive):

- **Reordered experimental presentation for improved narrative flow**. The revealed preferences experiments now appear first (Section 3.1), establishing what changes holistically before examining how robustly and coherently these changes manifest. This section includes revised experimental details (addressing reviewer pZBZ's concerns) and expanded results demonstrating that character training induces a similar pattern of strong preferences from different initial models.
- **Strengthened coherence evaluation through cross-judge replication** (Section 3.3). Following concerns from multiple reviewers (particularly yL1Y regarding potential self-bias), we now evaluate coherence using three frontier reasoning models (GPT-5 mini, Claude Haiku 4.5, Gemini 2.0 Flash-Lite) instead of a single judge. Results show consistent agreement across all three judges, with character training achieving consistently high win rates against activation steering across models.
- **New ablation studies on the introspection stage** (Appendix B.4). These experiments demonstrate that neither self-reflection nor self-interaction alone provides substantial gains over distillation only, but their combination is necessary for optimal robustness and coherence. Additional coherence comparisons with prompting and distillation-only baselines are now provided in Appendix D.

We believe these revisions significantly strengthen the paper and directly address the concerns raised during review.

---

### Meta-Review · Area_Chair_x1rh · 2026-01-12

**Summary:**

This paper proposes an open, end-to-end pipeline for character training of LLMs, combining constitution writing, DPO-based distillation from a teacher model, and an introspective SFT stage using self-generated data. Experiments across three open models and eleven personas suggest improved robustness to “break character” prompts, higher coherence than activation steering, and systematic shifts in revealed trait preferences.

The paper is clearly written, releases code, and explores an under-documented but practically relevant setting. Reviewers appreciated the breadth of personas and the attempt to go beyond prompting and steering (pZBZ, M1XF).

However, substantial weaknesses remain. Several reviewers highlight conceptual confusion around what constitutes a “persona,” mixing style, values, and beliefs without clear definitions, leading to self-contradictory framing (pZBZ). Methodologically, the approach relies heavily on hand-written constitutions and model-generated introspection data, raising scalability and circularity concerns. Most critically, the evaluation depends almost entirely on model-based judges and classifiers, without human validation, leaving the central claims about coherence, robustness, and behavioral change insufficiently grounded (M1XF, yL1Y, d6X5). Comparisons to other post-training baselines are also limited.

Overall, while promising, the contribution and evidence do not yet seem to meet ICLR’s bar for acceptance.

**Reviewer Concerns:**

The primary concern, shared by Reviewers M1XF, pZBZ, and yL1Y, is the heavy reliance on model-based evaluators. The lack of human validation for the "coherence" and "revealed-preference" metrics raises significant questions about whether the observed improvements reflect genuine behavioral shifts or merely alignment with the evaluator’s inductive biases. Furthermore, Reviewer pZBZ noted a lack of theoretical grounding in the definition of "personas," leading to a confusing framing that conflates style with moral values. Methodologically, the decision to perform DPO before SFT (Reviewer yL1Y) and the limited prompt diversity in the introspection stage (Reviewer d6X5) were viewed as weaknesses. Despite the authors' detailed rebuttals and additional ablations, the fundamental concerns regarding evaluation reliability and conceptual clarity remain.

**Reviewer Scores:**

It is hard to tell. The authors did provide ample responses to the reviewers' queries.

---

### Decision · Program_Chairs · 2026-01-26

Reject